# Concavity of reweighted Kikuchi approximation

**Po-Ling Loh**
Department of Statistics
The Wharton School
University of Pennsylvania
loh@wharton.upenn.edu

**Andre Wibisono**
Computer Science Division
University of California, Berkeley
wibisono@berkeley.edu

## Abstract

We analyze a reweighted version of the Kikuchi approximation for estimating the log partition function of a product distribution defined over a region graph. We establish sufficient conditions for the concavity of our reweighted objective function in terms of weight assignments in the Kikuchi expansion, and show that a reweighted version of the sum product algorithm applied to the Kikuchi region graph will produce global optima of the Kikuchi approximation whenever the algorithm converges. When the region graph has two layers, corresponding to a Bethe approximation, we show that our sufficient conditions for concavity are also necessary. Finally, we provide an explicit characterization of the polytope of concavity in terms of the cycle structure of the region graph. We conclude with simulations that demonstrate the advantages of the reweighted Kikuchi approach.

## 1 Introduction

Undirected graphical models are a familiar framework in diverse application domains such as computer vision, statistical physics, coding theory, social science, and epidemiology. In certain settings of interest, one is provided with potential functions defined over nodes and (hyper)edges of the graph. A crucial step in probabilistic inference is to compute the log partition function of the distribution based on these potential functions for a given graph structure. However, computing the log partition function either exactly or approximately is NP-hard in general [2, 17]. An active area of research involves finding accurate approximations of the log partition function and characterizing the graph structures for which such approximations may be computed efficiently [29, 22, 7, 19, 25, 18].

When the underlying graph is a tree, the log partition function may be computed exactly via the sum product algorithm in time linear in the number of nodes [15]. However, when the graph contains cycles, a generalized version of the sum product algorithm known as loopy belief propagation may either fail to converge or terminate in local optima of a nonconvex objective function [26, 20, 8, 13].

In this paper, we analyze the Kikuchi approximation method, which is constructed from a variational representation of the log partition function by replacing the entropy with an expression that decomposes with respect to a region graph. Kikuchi approximations were previously introduced in the physics literature [9] and reformalized by Yedidia et al. [28, 29] and others [1, 14] in the language of graphical models. The Bethe approximation, which is a special case of the Kikuchi approximation when the region graph has only two layers, has been studied by various authors [3, 28, 5, 25]. In addition, a reweighted version of the Bethe approximation was proposed by Wainwright et al. [22, 16]. As described in Vontobel [21], computing the global optimum of the Bethe variational problem may in turn be used to approximate the permanent of a nonnegative square matrix.

The particular objective function that we study generalizes the Kikuchi objective appearing in previous literature by assigning arbitrary weights to individual terms in the Kikuchi entropy expansion. We establish necessary and sufficient conditions under which this class of objective functions is concave, so a global optimum may be found efficiently. Our theoretical results synthesize known results on Kikuchi and Bethe approximations, and our main theorem concerning concavity conditions for the reweighted Kikuchi entropy recovers existing results when specialized to the unweighted

Kikuchi [14] or reweighted Bethe [22] case. Furthermore, we provide a valuable converse result in the reweighted Bethe case, showing that when our concavity conditions are violated, the entropy function cannot be concave over the whole feasible region. As demonstrated by our experiments, a message-passing algorithm designed to optimize the Kikuchi objective may terminate in local optima for weights outside the concave region. Watanabe and Fukumizu [24, 25] provide a similar converse in the unweighted Bethe case, but our proof is much simpler and our result is more general.

In the reweighted Bethe setting, we also present a useful characterization of the concave region of the Bethe entropy function in terms of the geometry of the graph. Specifically, we show that if the region graph consists of only singleton vertices and pairwise edges, then the region of concavity coincides with the convex hull of incidence vectors of single-cycle forest subgraphs of the original graph. When the region graph contains regions with cardinality greater than two, the latter region may be strictly contained in the former; however, our result provides a useful way to generate weight vectors within the region of concavity. Whereas Wainwright et al. [22] establish the concavity of the reweighted Bethe objective on the spanning forest polytope, that region is contained within the single-cycle forest polytope, and our simulations show that generating weight vectors in the latter polytope may yield closer approximations to the log partition function.

The remainder of the paper is organized as follows: In Section 2, we review background information about the Kikuchi and Bethe approximations. In Section 3, we provide our main results on concavity conditions for the reweighted Kikuchi approximation, including a geometric characterization of the region of concavity in the Bethe case. Section 4 outlines the reweighted sum product algorithm and proves that fixed points correspond to global optima of the Kikuchi approximation. Section 5 presents experiments showing the improved accuracy of the reweighted Kikuchi approximation over the region of concavity. Technical proofs and additional simulations are contained in the Appendix.

## 2 Background and problem setup

In this section, we review basic concepts of the Kikuchi approximation and establish some terminology to be used in the paper.

Let $G = (V, R)$ denote a *region graph* defined over the vertex set $V$, where each region $r \in R$ is a subset of $V$. Directed edges correspond to inclusion, so $r \to s$ is an edge of $G$ if $s \subseteq r$. We use the following notation, for $r \in R$:

$$
\begin{aligned}
\mathcal{A}(r) &:= \{s \in R \colon r \subsetneq s\} && (\textit{ancestors of } r) \\
\mathcal{F}(r) &:= \{s \in R \colon r \subseteq s\} && (\textit{forebears of } r) \\
N(r) &:= \{s \in R \colon r \subseteq s \text{ or } s \subseteq r\} && (\textit{neighbors of } r).
\end{aligned}
$$

For $R' \subseteq R$, we define $\mathcal{A}(R') = \bigcup_{r \in R'} \mathcal{A}(r)$, and we define $\mathcal{F}(R')$ and $N(R')$ similarly.

We consider joint distributions $x = (x_s)_{s \in V}$ that factorize over the region graph; i.e.,

$$
p(x) = \frac{1}{Z(\alpha)} \prod_{r \in R} \alpha_r(x_r), \tag{1}
$$

for potential functions $\alpha_r > 0$. Here, $Z(\alpha)$ is the normalization factor, or partition function, which is a function of the potential functions $\alpha_r$, and each variable $x_s$ takes values in a finite discrete set $\mathcal{X}$. One special case of the factorization (1) is the pairwise Ising model, defined over a graph $G = (V, E)$, where the distribution is given by

$$
p_\gamma(x) = \exp\Big( \sum_{s \in V} \gamma_s(x_s) + \sum_{(s,t) \in E} \gamma_{st}(x_s, x_t) - A(\gamma) \Big), \tag{2}
$$

and $\mathcal{X} = \{-1, +1\}$. Our goal is to analyze the log partition function

$$
\log Z(\alpha) = \log \Big\{ \sum_{x \in \mathcal{X}^{|V|}} \prod_{r \in R} \alpha_r(x_r) \Big\}. \tag{3}
$$

### 2.1 Variational representation

It is known from the theory of graphical models [14] that the log partition function (3) may be written in the variational form

$$
\log Z(\alpha) = \sup_{\{\tau_r(x_r)\} \in \Delta_R} \Big\{ \sum_{r \in R} \sum_{x_r} \tau_r(x_r) \log(\alpha_r(x_r)) + H(p_\tau) \Big\}, \tag{4}
$$

where $p_\tau$ is the maximum entropy distribution with marginals $\{\tau_r(x_r)\}$ and

$$H(p) := -\sum_x p(x) \log p(x)$$

is the usual entropy. Here, $\Delta_R$ denotes the $R$-marginal polytope; i.e., $\{\tau_r(x_r) \colon r \in R\} \in \Delta_R$ if and only if there exists a distribution $\tau(x)$ such that $\tau_r(x_r) = \sum_{x_{\backslash r}} \tau(x_r, x_{\backslash r})$ for all $r$. For ease of notation, we also write $\tau \equiv \{\tau_r(x_r) \colon r \in R\}$. Let $\theta \equiv \theta(x)$ denote the collection of log potential functions $\{\log(\alpha_r(x_r)) \colon r \in R\}$. Then equation (4) may be rewritten as

$$\log Z(\theta) = \sup_{\tau \in \Delta_R} \{\langle \theta, \tau \rangle + H(p_\tau)\}. \tag{5}$$

Specializing to the Ising model (2), equation (5) gives the variational representation

$$A(\gamma) = \sup_{\mu \in \mathbb{M}} \{\langle \gamma, \mu \rangle + H(p_\mu)\}, \tag{6}$$

which appears in Wainwright and Jordan [23]. Here, $\mathbb{M} \equiv \mathbb{M}(G)$ denotes the marginal polytope, corresponding to the collection of mean parameter vectors of the sufficient statistics in the exponential family representation (2), ranging over different values of $\gamma$, and $p_\mu$ is the maximum entropy distribution with mean parameters $\mu$.

## 2.2 Reweighted Kikuchi approximation

Although the set $\Delta_R$ appearing in the variational representation (5) is a convex polytope, it may have exponentially many facets [23]. Hence, we replace $\Delta_R$ with the set

$$\Delta_R^K = \left\{ \tau \colon \forall t, u \in R \text{ s.t. } t \subseteq u, \sum_{x_{u \backslash t}} \tau_u(x_t, x_{u \backslash t}) = \tau_t(x_t) \quad \text{and} \quad \forall u \in R, \sum_{x_u} \tau_u(x_u) = 1 \right\}$$

of *locally consistent $R$-pseudomarginals*. Note that $\Delta_R \subseteq \Delta_R^K$ and the latter set has only polynomially many facets, making optimization more tractable.

In the case of the pairwise Ising model (2), we let $\mathbb{L} \equiv \mathbb{L}(G)$ denote the polytope $\Delta_R^K$. Then $\mathbb{L}$ is the collection of nonnegative functions $\tau = (\tau_s, \tau_{st})$ satisfying the marginalization constraints

$$\sum_{x_s} \tau_s(x_s) = 1, \qquad\qquad \forall s \in V,$$
$$\sum_{x_t} \tau_{st}(x_s, x_t) = \tau_s(x_s) \text{ and } \sum_{x_s} \tau_{st}(x_s, x_t) = \tau_t(x_t), \qquad \forall (s,t) \in E.$$

Recall that $\mathbb{M}(G) \subseteq \mathbb{L}(G)$, with equality achieved if and only if the underlying graph $G$ is a tree. In the general case, we have $\Delta_R = \Delta_R^K$ when the Hasse diagram of the region graph admits a minimal representation that is loop-free (cf. Theorem 2 of Pakzad and Anantharam [14]).

Given a collection of $R$-pseudomarginals $\tau$, we also replace the entropy term $H(p_\tau)$, which is difficult to compute in general, by the approximation

$$H(p_\tau) \approx \sum_{r \in R} \rho_r H_r(\tau_r) := H(\tau; \rho), \tag{7}$$

where $H_r(\tau_r) := -\sum_{x_r} \tau_r(x_r) \log \tau_r(x_r)$ is the entropy computed over region $r$, and $\{\rho_r \colon r \in R\}$ are weights assigned to the regions. Note that in the pairwise Ising case (2), with $p := p_\gamma$, we have the equality

$$H(p) = \sum_{s \in V} H_s(p_s) - \sum_{(s,t) \in E} I_{st}(p_{st})$$

when $G$ is a tree, where $I_{st}(p_{st}) = H_s(p_s) + H_t(p_t) - H_{st}(p_{st})$ denotes the mutual information and $p_s$ and $p_{st}$ denote the node and edge marginals. Hence, the approximation (7) is exact with

$$\rho_{st} = 1, \quad \forall (s,t) \in E, \qquad \text{and} \qquad \rho_s = 1 - \deg(s), \quad \forall s \in V.$$

Using the approximation (7), we arrive at the following *reweighted Kikuchi approximation*:

$$B(\theta; \rho) := \sup_{\tau \in \Delta_R^K} \underbrace{\{\langle \theta, \tau \rangle + H(\tau; \rho)\}}_{B_{\theta, \rho}(\tau)}. \tag{8}$$

Note that when $\{\rho_r\}$ are the *overcounting numbers* $\{c_r\}$, defined recursively by

$$c_r = 1 - \sum_{s \in \mathcal{A}(r)} c_s, \tag{9}$$

the expression (8) reduces to the usual (unweighted) Kikuchi approximation considered in Pakzad and Anantharam [14].

# 3 Main results and consequences

In this section, we analyze the concavity of the Kikuchi variational problem (8). We derive a sufficient condition under which the function $B_{\theta,\rho}(\tau)$ is concave over the set $\Delta_R^K$, so global optima of the reweighted Kikuchi approximation may be found efficiently. In the Bethe case, we also show that the condition is necessary for $B_{\theta,\rho}(\tau)$ to be concave over the entire region $\Delta_R^K$, and we provide a geometric characterization of $\Delta_R^K$ in terms of the edge and cycle structure of the graph.

## 3.1 Sufficient conditions for concavity

We begin by establishing sufficient conditions for the concavity of $B_{\theta,\rho}(\tau)$. Clearly, this is equivalent to establishing conditions under which $H(\tau;\rho)$ is concave. Our main result is the following:

**Theorem 1.** *If $\rho \in \mathbb{R}^{|R|}$ satisfies*

$$\sum_{s \in \mathcal{F}(S)} \rho_s \geq 0, \qquad \forall S \subseteq R, \tag{10}$$

*then the Kikuchi entropy $H(\tau;\rho)$ is strictly concave on $\Delta_R^K$.*

The proof of Theorem 1 is contained in Appendix A.1, and makes use of a generalization of Hall's marriage lemma for weighted graphs (cf. Lemma 1 in Appendix A.2).

The condition (10) depends heavily on the structure of the region graph. For the sake of interpretability, we now specialize to the case where the region graph has only two layers, with the first layer corresponding to vertices and the second layer corresponding to hyperedges. In other words, for $r, s \in R$, we have $r \subseteq s$ only if $|r| = 1$, and $R = V \cup F$, where $F$ is the set of hyperedges and $V$ denotes the set of singleton vertices. This is the *Bethe case*, and the entropy

$$H(\tau;\rho) = \sum_{s \in V} \rho_s H_s(\tau_s) + \sum_{\alpha \in F} \rho_\alpha H_\alpha(\tau_\alpha) \tag{11}$$

is consequently known as the Bethe entropy.

The following result is proved in Appendix A.3:

**Corollary 1.** *Suppose $\rho_\alpha \geq 0$ for all $\alpha \in F$, and the following condition also holds:*

$$\sum_{s \in U} \rho_s + \sum_{\alpha \in F : \, \alpha \cap U \neq \emptyset} \rho_\alpha \geq 0, \qquad \forall U \subseteq V. \tag{12}$$

*Then the Bethe entropy $H(\tau;\rho)$ is strictly concave over $\Delta_R^K$.*

## 3.2 Necessary conditions for concavity

We now establish a converse to Corollary 1 in the Bethe case, showing that condition (12) is also necessary for the concavity of the Bethe entropy. When $\rho_\alpha = 1$ for $\alpha \in F$ and $\rho_s = 1 - |N(s)|$ for $s \in V$, we recover the result of Watanabe and Fukumizu [25] for the unweighted Bethe case. However, our proof technique is significantly simpler and avoids the complex machinery of graph zeta functions. Our approach proceeds by considering the Bethe entropy $H(\tau;\rho)$ on appropriate slices of the domain $\Delta_R^K$ so as to extract condition (12) for each $U \subseteq V$. The full proof is provided in Appendix B.1.

**Theorem 2.** *If the Bethe entropy $H(\tau;\rho)$ is concave over $\Delta_R^K$, then $\rho_\alpha \geq 0$ for all $\alpha \in F$, and condition (12) holds.*

Indeed, as demonstrated in the simulations of Section 5, the Bethe objective function $B_{\theta,\rho}(\tau)$ may have multiple local optima if $\rho$ does *not* satisfy condition (12).

## 3.3 Polytope of concavity

We now characterize the polytope defined by the inequalities (12). We show that in the pairwise Bethe case, the polytope may be expressed geometrically as the convex hull of single-cycle forests

formed by the edges of the graph. In the more general (non-pairwise) Bethe case, however, the polytope of concavity may strictly contain the latter set.

Note that the Bethe entropy (11) may be written in the alternative form

$$H(\tau; \rho) = \sum_{s \in V} \rho'_s H_s(\tau_s) - \sum_{\alpha \in F} \rho_\alpha \widetilde{I}_\alpha(\tau_\alpha), \tag{13}$$

where $\widetilde{I}_\alpha(\tau_\alpha) := \{\sum_{s \in \alpha} H_s(\tau_s)\} - H_\alpha(\tau_\alpha)$ is the KL divergence between the joint distribution $\tau_\alpha$ and the product distribution $\prod_{s \in \alpha} \tau_s$, and the weights $\rho'_s$ are defined appropriately.

We show that the polytope of concavity has a nice geometric characterization when $\rho'_s = 1$ for all $s \in V$, and $\rho_\alpha \in [0, 1]$ for all $\alpha \in F$. Note that this assignment produces the expression for the reweighted Bethe entropy analyzed in Wainwright et al. [22] (when all elements of $F$ have cardinality two). Equation (13) then becomes

$$H(\tau; \rho) = \sum_{s \in V} \Big(1 - \sum_{\alpha \in N(s)} \rho_\alpha\Big) H_s(\tau_s) + \sum_{\alpha \in F} \rho_\alpha H_\alpha(\tau_\alpha), \tag{14}$$

and the inequalities (12) defining the polytope of concavity are

$$\sum_{\alpha \in F: \, \alpha \cap U \neq \emptyset} (|\alpha \cap U| - 1)\rho_\alpha \leq |U|, \quad \forall U \subseteq V. \tag{15}$$

Consequently, we define

$$\mathbb{C} := \Big\{\rho \in [0, 1]^{|F|} : \sum_{\alpha \in F: \, \alpha \cap U \neq \emptyset} (|\alpha \cap U| - 1)\rho_\alpha \leq |U|, \quad \forall U \subseteq V\Big\}.$$

By Theorem 2, the set $\mathbb{C}$ is the region of concavity for the Bethe entropy (14) within $[0, 1]^{|F|}$.

We also define the set

$$\mathbb{F} := \{1_{F'} : F' \subseteq F \text{ and } F' \cup N(F') \text{ is a single-cycle forest in } G\} \subseteq \{0, 1\}^{|F|},$$

where a *single-cycle forest* is defined to be a subset of edges of a graph such that each connected component contains at most one cycle. (We disregard the directions of edges in $G$.) The following theorem gives our main result. The proof is contained in Appendix C.1.

**Theorem 3.** *In the Bethe case (i.e., the region graph $G$ has two layers), we have the containment* $\mathrm{conv}(\mathbb{F}) \subseteq \mathbb{C}$. *If in addition $|\alpha| = 2$ for all $\alpha \in F$, then $\mathrm{conv}(\mathbb{F}) = \mathbb{C}$.*

The significance of Theorem 3 is that it provides us with a convenient graph-based method for constructing vectors $\rho \in \mathbb{C}$. From the inequalities (15), it is not even clear how to efficiently verify whether a given $\rho \in [0, 1]^{|F|}$ lies in $\mathbb{C}$, since it involves testing $2^{|V|}$ inequalities.

Comparing Theorem 3 with known results, note that in the pairwise case ($|\alpha| = 2$ for all $\alpha \in F$), Theorem 1 of Wainwright et al. [22] states that the Bethe entropy is concave over $\mathrm{conv}(\mathbb{T})$, where $\mathbb{T} \subseteq \{0, 1\}^{|E|}$ is the set of edge indicator vectors for spanning forests of the graph. It is trivial to check that $\mathbb{T} \subseteq \mathbb{F}$, since every spanning forest is also a single-cycle forest. Hence, Theorems 2 and 3 together imply a stronger result than in Wainwright et al. [22], characterizing the precise region of concavity for the Bethe entropy as a superset of the polytope $\mathrm{conv}(\mathbb{T})$ analyzed there. In the unweighted Kikuchi case, it is also known [1, 14] that the Kikuchi entropy is concave for the assignment $\rho = 1_F$ when the region graph $G$ is connected and has at most one cycle. Clearly, $1_F \in \mathbb{C}$ in that case, so this result is a consequence of Theorems 2 and 3, as well. However, our theorems show that a much more general statement is true.

It is tempting to posit that $\mathrm{conv}(\mathbb{F}) = \mathbb{C}$ holds more generally in the Bethe case. However, as the following example shows, settings arise where $\mathrm{conv}(\mathbb{F}) \subsetneq \mathbb{C}$. Details are contained in Appendix C.2.

**Example 1.** Consider a two-layer region graph with vertices $V = \{1, 2, 3, 4, 5\}$ and factors $\alpha_1 = \{1, 2, 3\}$, $\alpha_2 = \{2, 3, 4\}$, and $\alpha_3 = \{3, 4, 5\}$. Then $(1, \frac{1}{2}, 1) \in \mathbb{C} \backslash \mathrm{conv}(\mathbb{F})$.

In fact, Example 1 is a special case of a more general statement, which we state in the following proposition. Here, $\mathfrak{F} := \{F' \subseteq F : 1_{F'} \in \mathbb{F}\}$, and an element $F^* \in \mathfrak{F}$ is *maximal* if it is not contained in another element of $\mathfrak{F}$.

**Proposition 1.** *Suppose (i) $G$ is not a single-cycle forest, and (ii) there exists a maximal element $F^* \in \mathfrak{F}$ such that the induced subgraph $F^* \cup N(F^*)$ is a forest. Then $conv(\mathbb{F}) \subsetneq \mathbb{C}$.*

The proof of Proposition 1 is contained in Appendix C.3. Note that if $|\alpha| = 2$ for all $\alpha \in F$, then condition (ii) is violated whenever condition (i) holds, so Proposition 1 provides a partial converse to Theorem 3.

## 4  Reweighted sum product algorithm

In this section, we provide an iterative message passing algorithm to optimize the Kikuchi variational problem (8). As in the case of the generalized belief propagation algorithm for the unweighted Kikuchi approximation [28, 29, 11, 14, 12, 27] and the reweighted sum product algorithm for the Bethe approximation [22], our message passing algorithm searches for stationary points of the Lagrangian version of the problem (8). When $\rho$ satisfies condition (10), Theorem 1 implies that the problem (8) is strictly concave, so the unique fixed point of the message passing algorithm globally maximizes the Kikuchi approximation.

Let $G = (V, R)$ be a region graph defining our Kikuchi approximation. Following Pakzad and Anantharam [14], for $r, s \in R$, we write $r \prec s$ if $r \subsetneq s$ and there does not exist $t \in R$ such that $r \subsetneq t \subsetneq s$. For $r \in R$, we define the parent set of $r$ to be $\mathcal{P}(r) = \{s \in R \colon r \prec s\}$ and the child set of $r$ to be $\mathcal{C}(r) = \{s \in R \colon s \prec r\}$. With this notation, $\tau = \{\tau_r(x_r) \colon r \in R\}$ belongs to the set $\Delta_R^K$ if and only if $\sum_{x_{s \setminus r}} \tau_s(x_r, x_{s \setminus r}) = \tau_r(x_r)$ for all $r \in R$, $s \in \mathcal{P}(r)$.

The message passing algorithm we propose is as follows: For each $r \in R$ and $s \in \mathcal{P}(r)$, let $M_{sr}(x_r)$ denote the message passed from $s$ to $r$ at assignment $x_r$. Starting with an arbitrary positive initialization of the messages, we repeatedly perform the following updates for all $r \in R$, $s \in \mathcal{P}(r)$:

$$M_{sr}(x_r) \leftarrow C \left[ \frac{\sum_{x_{s \setminus r}} \exp\left(\theta_s(x_s)/\rho_s\right) \prod_{v \in \mathcal{P}(s)} M_{vs}(x_s)^{\rho_v/\rho_s} \prod_{w \in \mathcal{C}(s) \setminus r} M_{sw}(x_w)^{-1}}{\exp\left(\theta_r(x_r)/\rho_r\right) \prod_{u \in \mathcal{P}(r) \setminus s} M_{ur}(x_r)^{\rho_u/\rho_r} \prod_{t \in \mathcal{C}(r)} M_{rt}(x_t)^{-1}} \right]^{\frac{\rho_r}{\rho_r + \rho_s}}. \quad (16)$$

Here, $C > 0$ may be chosen to ensure a convenient normalization condition; e.g., $\sum_{x_r} M_{sr}(x_r) = 1$. Upon convergence of the updates (16), we compute the pseudomarginals according to

$$\tau_r(x_r) \propto \exp\left(\frac{\theta_r(x_r)}{\rho_r}\right) \prod_{s \in \mathcal{P}(r)} M_{sr}(x_r)^{\rho_s/\rho_r} \prod_{t \in \mathcal{C}(r)} M_{rt}(x_t)^{-1}, \quad (17)$$

and we obtain the corresponding Kikuchi approximation by computing the objective function (8) with these pseudomarginals. We have the following result, which is proved in Appendix D:

**Theorem 4.** *The pseudomarginals $\tau$ specified by the fixed points of the messages $\{M_{sr}(x_r)\}$ via the updates (16) and (17) correspond to the stationary points of the Lagrangian associated with the Kikuchi approximation problem (8).*

As with the standard belief propagation and reweighted sum product algorithms, we have several options for implementing the above message passing algorithm in practice. For example, we may perform the updates (16) using serial or parallel schedules. To improve the convergence of the algorithm, we may damp the updates by taking a convex combination of new and previous messages using an appropriately chosen step size. As noted by Pakzad and Anantharam [14], we may also use a minimal graphical representation of the Hasse diagram to lower the complexity of the algorithm.

Finally, we remark that although our message passing algorithm proceeds in the same spirit as classical belief propagation algorithms by operating on the Lagrangian of the objective function, our algorithm as presented above does not immediately reduce to the generalized belief propagation algorithm for unweighted Kikuchi approximations or the reweighted sum product algorithm for tree-reweighted pairwise Bethe approximations. Previous authors use algebraic relations between the overcounting numbers (9) in the Kikuchi case [28, 29, 11, 14] and the two-layer structure of the Hasse diagram in the Bethe case [22] to obtain a simplified form of the updates. Since the coefficients $\rho$ in our problem lack the same algebraic relations, following the message-passing protocol used in previous work [11, 28] leads to more complicated updates, so we present a slightly different algorithm that still optimizes the general reweighted Kikuchi objective.

# 5 Experiments

In this section, we present empirical results to demonstrate the advantages of the reweighted Kikuchi approximation that support our theoretical results. For simplicity, we focus on the binary pairwise Ising model given in equation (2). Without loss of generality, we may take the potentials to be $\gamma_s(x_s) = \gamma_s x_s$ and $\gamma_{st}(x_s, x_t) = \gamma_{st} x_s x_t$ for some $\gamma = (\gamma_s, \gamma_{st}) \in \mathbb{R}^{|V|+|E|}$. We run our experiments on two types of graphs: (1) $K_n$, the complete graph on $n$ vertices, and (2) $T_n$, the $\sqrt{n} \times \sqrt{n}$ toroidal grid graph where every vertex has degree four.

**Bethe approximation.** We consider the pairwise Bethe approximation of the log partition function $A(\gamma)$ with weights $\rho_{st} \geq 0$ and $\rho_s = 1 - \sum_{t \in N(s)} \rho_{st}$. Because of the regularity structure of $K_n$ and $T_n$, we take $\rho_{st} = \rho \geq 0$ for all $(s, t) \in E$ and study the behavior of the Bethe approximation as $\rho$ varies. For this particular choice of weight vector $\vec{\rho} = \rho 1_E$, we define

$$\rho_{\text{tree}} = \max\{\rho \geq 0 \colon \vec{\rho} \in \text{conv}(\mathbb{T})\}, \qquad \text{and} \qquad \rho_{\text{cycle}} = \max\{\rho \geq 0 \colon \vec{\rho} \in \text{conv}(\mathbb{F})\}.$$

It is easily verified that for $K_n$, we have $\rho_{\text{tree}} = \frac{2}{n}$ and $\rho_{\text{cycle}} = \frac{2}{n-1}$; while for $T_n$, we have $\rho_{\text{tree}} = \frac{n-1}{2n}$ and $\rho_{\text{cycle}} = \frac{1}{2}$.

Our results in Section 3 imply that the Bethe objective function $B_{\gamma,\rho}(\tau)$ in equation (8) is concave if and only if $\rho \leq \rho_{\text{cycle}}$, and Wainwright et al. [22] show that we have the bound $A(\gamma) \leq B(\gamma; \rho)$ for $\rho \leq \rho_{\text{tree}}$. Moreover, since the Bethe entropy may be written in terms of the edge mutual information (13), the function $B(\gamma; \rho)$ is decreasing in $\rho$. In our results below, we observe that we may obtain a tighter approximation to $A(\gamma)$ by moving from the upper bound region $\rho \leq \rho_{\text{tree}}$ to the concavity region $\rho \leq \rho_{\text{cycle}}$. In addition, for $\rho > \rho_{\text{cycle}}$, we observe multiple local optima of $B_{\gamma,\rho}(\tau)$.

**Procedure.** We generate a random potential $\gamma = (\gamma_s, \gamma_{st}) \in \mathbb{R}^{|V|+|E|}$ for the Ising model (2) by sampling each potential $\{\gamma_s\}_{s \in V}$ and $\{\gamma_{st}\}_{(s,t) \in E}$ independently. We consider two types of models:

$$\textit{Attractive}: \ \gamma_{st} \sim \text{Uniform}[0, \omega_{st}], \qquad \text{and} \qquad \textit{Mixed}: \ \gamma_{st} \sim \text{Uniform}[-\omega_{st}, \omega_{st}].$$

In each case, $\gamma_s \sim \text{Uniform}[0, \omega_s]$. We set $\omega_s = 0.1$ and $\omega_{st} = 2$. Intuitively, the attractive model encourages variables in adjacent nodes to assume the same value, and it has been shown [18, 19] that the ordinary Bethe approximation ($\rho_{st} = 1$) in an attractive model lower-bounds the log partition function. For $\rho \in [0, 2]$, we compute stationary points of $B_{\gamma,\rho}(\tau)$ by running the reweighted sum product algorithm of Wainwright et al. [22]. We use a damping factor of $\lambda = 0.5$, convergence threshold of $10^{-10}$ for the average change of messages, and at most 2500 iterations. We repeat this process with at least 8 random initializations for each value of $\rho$. Figure 1 shows the scatter plots of $\rho$ and the Bethe approximation $B_{\gamma,\rho}(\tau)$. In each plot, the two vertical lines are the boundaries $\rho = \rho_{\text{tree}}$ and $\rho = \rho_{\text{cycle}}$, and the horizontal line is the value of the true log partition function $A(\gamma)$.

**Results.** Figures 1(a)–1(d) show the results of our experiments on small graphs ($K_5$ and $T_9$) for both attractive and mixed models. We see that the Bethe approximation with $\rho \leq \rho_{\text{cycle}}$ generally provides a better approximation to $A(\gamma)$ than the Bethe approximation computed over $\rho \leq \rho_{\text{tree}}$. However, in general we cannot guarantee whether $B(\gamma; \rho)$ will give an upper or lower bound for $A(\gamma)$ when $\rho \leq \rho_{\text{cycle}}$. As noted above, we have $B(\gamma; 1) \leq A(\gamma)$ for attractive models.

We also observe from Figures 1(a)–1(d) that shortly after $\rho$ leaves the concavity region $\{\rho \leq \rho_{\text{cycle}}\}$, multiple local optima emerge for the Bethe objective function. The presence of the point clouds near $\rho = 1$ in Figures 1(a) and 1(c) arises because the sum product algorithm has not converged after 2500 iterations. Indeed, the same phenomenon is true for all our results: in the region where multiple local optima begin to appear, it is more difficult for the algorithm to converge. See Figure 2 and the accompanying text in Appendix E for a plot of the points $(\rho, \log_{10}(\Delta))$, where $\Delta$ is the final average change in the messages at termination of the algorithm. From Figure 2, we see that the values of $\Delta$ are significantly higher for the values of $\rho$ near where multiple local optima emerge. We suspect that for these values of $\rho$, the sum product algorithm fails to converge since distinct local optima are close together, so messages oscillate between the optima. For larger values of $\rho$, the local optima become sufficiently separated and the algorithm converges to one of them. However, it is interesting to note that this point cloud phenomenon does not appear for attractive models, despite the presence of distinct local optima.

Simulations for larger graphs are shown in Figures 1(e)–1(h). If we zoom into the region near $\rho \leq \rho_{\text{cycle}}$, we still observe the same behavior that $\rho \leq \rho_{\text{cycle}}$ generally provides a better Bethe

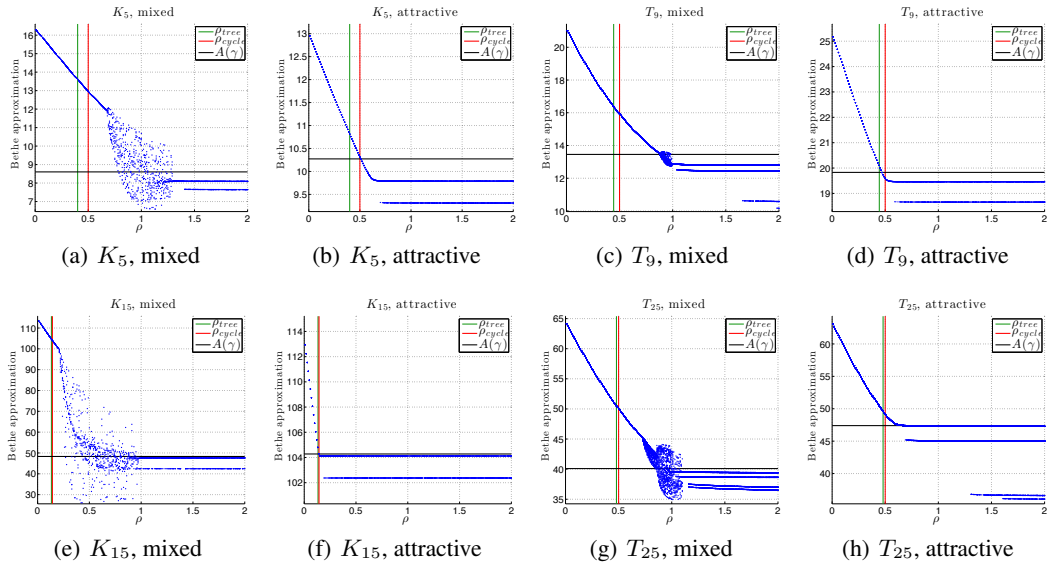

| | | | |
|---|---|---|---|
| (a) $K_5$, mixed | (b) $K_5$, attractive | (c) $T_9$, mixed | (d) $T_9$, attractive |
| (e) $K_{15}$, mixed | (f) $K_{15}$, attractive | (g) $T_{25}$, mixed | (h) $T_{25}$, attractive |

**Figure 1:** Values of the reweighted Bethe approximation as a function of $\rho$. See text for details.

approximation than $\rho \leq \rho_{\text{tree}}$. Moreover, the presence of the point clouds and multiple local optima are more pronounced, and we see from Figures 1(c), 1(g), and 1(h) that new local optima with even worse Bethe values arise for larger values of $\rho$. Finally, we note that the same qualitative behavior also occurs in all the other graphs that we have tried ($K_n$ for $n \in \{5, 10, 15, 20, 25\}$ and $T_n$ for $n \in \{9, 16, 25, 36, 49, 64\}$), with multiple random instances of the Ising model $p_\gamma$.

## 6 Discussion

In this paper, we have analyzed the reweighted Kikuchi approximation method for estimating the log partition function of a distribution that factorizes over a region graph. We have characterized necessary and sufficient conditions for the concavity of the variational objective function, generalizing existing results in literature. Our simulations demonstrate the advantages of using the reweighted Kikuchi approximation and show that multiple local optima may appear outside the region of concavity.

An interesting future research direction is to obtain a better understanding of the approximation guarantees of the reweighted Bethe and Kikuchi methods. In the Bethe case with attractive potentials $\theta$, several recent results [22, 19, 18] establish that the Bethe approximation $B(\theta; \rho)$ is an upper bound to the log partition function $A(\theta)$ when $\rho$ lies in the spanning tree polytope, whereas $B(\theta; \rho) \leq A(\theta)$ when $\rho = 1_F$. By continuity, we must have $B(\theta; \rho^*) = A(\theta)$ for some values of $\rho^*$, and it would be interesting to characterize such values where the reweighted Bethe approximation is exact.

Another interesting direction is to extend our theoretical results on properties of the reweighted Kikuchi approximation, which currently depend solely on the structure of the region graph and the weights $\rho$, to incorporate the effect of the model potentials $\theta$. For example, several authors [20, 6] present conditions under which loopy belief propagation applied to the unweighted Bethe approximation has a unique fixed point. The conditions for uniqueness of fixed points slightly generalize the conditions for convexity, and they involve both the graph structure and the strength of the potentials. We suspect that similar results would hold for the reweighted Kikuchi approximation.

**Acknowledgments.** The authors thank Martin Wainwright for introducing the problem to them and providing helpful guidance. The authors also thank Varun Jog for discussions regarding the generalization of Hall's lemma. The authors thank the anonymous reviewers for feedback that improved the clarity of the paper. PL was partly supported from a Hertz Foundation Fellowship and an NSF Graduate Research Fellowship while at Berkeley.

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
