[Supplementary Material]

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

# A  Proofs for Section 3.1

## A.1  Proof of Theorem 1

We use the proof technique of Theorem 1 in Pakzad and Anantharam [14] for the unweighted Bethe entropy, together with Lemma 1 in Appendix A.2, which provides a generalization of Hall's marriage lemma for weighted bipartite graphs.

We construct a bipartite graph according to

$$V_1 := \{r \in R \colon \rho_r < 0\}, \quad \text{and} \quad V_2 := \{r \in R \colon \rho_r > 0\},$$

where $(s, t) \in E$ for $s \in V_1$ and $t \in V_2$ when $s \subset t$. Let weights $w$ be defined such that $w(s) = -\rho_s$ for $s \in V_1$ and $w(s) = \rho_s$ for $s \in V_2$. We claim that condition (19) of Lemma 1 is satisfied. Indeed, for $U \subseteq V_1$, we have

$$w(U) = -\sum_{s \in U} \rho_s \leq \sum_{s \in A(U)} \rho_s = \sum_{s \in A(U):\rho_s>0} \rho_s + \sum_{s \in A(U):\rho_s<0} \rho_s \leq \sum_{s \in A(U):\rho_s>0} \rho_s = w(N(U)),$$

where the first inequality is a direct application of the assumption (10). Hence, by Lemma 1, we have a saturating edge labeling $\gamma$.

For each $t \in V_2$, define

$$\rho'_t := \rho_t - \sum_{s \in N(t)} \gamma_{st} \geq 0.$$

We may then write

$$
\begin{aligned}
H(\tau; \rho) &= \sum_{s \in V_1} \rho_s H_s(\tau_s) + \sum_{t \in V_2} \rho_t H_t(\tau_t) \\
&= \sum_{(s,t) \in E} \gamma_{st} \left\{ -H_s(\tau_s) + H_t(\tau_t) \right\} + \sum_{t \in V_2} \rho'_t H_t(\tau_t) \\
&= \sum_{(s,t) \in E} \gamma_{st} \left\{ \sum_{x_s} \tau_s(x_s) \log \tau_s(x_s) - \sum_{x_t} \tau_t(x_t) \log \tau_t(x_t) \right\} + \sum_{t \in V_2} \rho'_t H_t(\tau_t) \\
&= \sum_{(s,t) \in E} \gamma_{st} \sum_{x_t} \tau_t(x_t) \log \left( \frac{\tau_s(x_s)}{\tau_t(x_t)} \right) + \sum_{t \in V_2} \rho'_t H_t(\tau_t), \quad\quad (18)
\end{aligned}
$$

where we have used the fact that $\sum_{x_{t\setminus s}} \tau_t(x_s, x_{t\setminus s}) = \tau_s(x_s)$, since $\tau \in \Delta_R^K$, to obtain the last equality.

Note that for each pair $(s, t)$, we have

$$\sum_{x_t} \tau_t(x_t) \log \left( \frac{\tau_s(x_s)}{\tau_t(x_t)} \right) = -D_{\mathrm{KL}}(\tau_t \| \tau_s),$$

which is strictly concave in the pair $(\tau_t, \tau_s)$. Furthermore, each term $H_t(\tau_t)$ is concave in $\tau_t$. It follows by the expansion (18) that $H(\tau; \rho)$ is strictly concave, as wanted.

## A.2  Generalization of Hall's marriage lemma

In this section, we prove a generalization of Hall's marriage lemma, which is useful in proving concavity of the Bethe entropy function $H(\tau; \rho)$.

Let $G = (V_1 \cup V_2, E)$ be a bipartite graph, where each vertex $v \in V := V_1 \cup V_2$ is assigned a weight $w(v) > 0$. For a set $U \subseteq V$, define

$$w(U) := \sum_{s \in U} w(s).$$

Also define the neighborhood set

$$N(U) := \bigcup_{s \in U} N(s),$$

where $N(s) := \{t \colon (s,t) \in E\}$ is the usual neighborhood set of a single node.

We say that an edge labeling $\gamma = (\gamma_{st} \colon (s,t) \in E) \in \mathbb{R}_{\geq 0}^{|E|}$ *saturates* $V_1$ if the following conditions hold:

1. For all $s \in V_1$, we have $\sum_{t \in N(s)} \gamma_{st} = w(s)$.
2. For all $t \in V_2$, we have $\sum_{s \in N(t)} \gamma_{st} \leq w(t)$.

**Lemma 1.** *Suppose*

$$w(U) \leq w(N(U)), \quad \forall U \subseteq V_1. \tag{19}$$

*Then there exists an edge labeling $\gamma$ that saturates $V_1$.*

*Proof.* We prove the lemma in stages. First, assume $w(v) \in \mathbb{Q}$ for all $v \in V$ and condition (19) holds. With an appropriate rescaling, we may assume that all weights are integers. Call the new weights $w'$. We then construct a graph $G'$ such that each node $v \in V$ is expanded into a set $U_v$ of $w'(v)$ nodes, and edges of $G'$ are constructed by connecting all nodes in $U_s$ to all nodes in $U_t$, for each $(s,t) \in E$. By the usual version of Hall's marriage lemma [4], there exists a matching of $G'$ that saturates $V_1' := \bigcup_{v \in V_1} U_v$. Indeed, it follows immediately from condition (19) that

$$w'(U) \leq w'(N(U)), \qquad \forall U \subseteq V_1.$$

Suppose $T' \subseteq V_1'$, and let $T := \{s \in V_1 \colon U_s \cap T' \neq \emptyset\}$. Then

$$|T'| \leq \left| \bigcup_{s \in T} U_s \right| = w'(T) \leq w'(N(T)) = |N(T')|,$$

so the sufficient condition of Hall's marriage lemma is met, implying the existence of a matching. The edge labeling $\gamma$ is obtained by setting

$$\gamma_{st} = \{\# \text{ of edges between } U_s \text{ and } U_t \text{ in matching}\}$$

and rescaling.

Next, suppose $w(v) \in \mathbb{R}$ for all $v \in V$ and condition (19) holds with *strict* inequality; i.e.,

$$w(U) < w(N(U)), \qquad \forall U \subseteq V_1. \tag{20}$$

We claim that there exists an edge labeling $\gamma$ that saturates $V_1$. Indeed, let

$$\epsilon := \min_{U \subseteq V_1} \{w(N(U)) - w(U)\} > 0.$$

Define a new weighting $w'$ with only rational values, such that

$$w'(s) \in \left[ w(s), \quad w(s) + \frac{\epsilon}{2 \cdot \deg(G)} \right), \qquad \forall s \in V_1,$$

$$w'(t) \in \left( w(t) - \frac{\epsilon}{2 \cdot \deg(G)}, \quad w(t) \right], \qquad \forall t \in V_2,$$

where $\deg(G) = |E|$ is the number of edges in $G$. It is clear that Hall's condition (19) still holds for $w'$. Hence, by the result of the last paragraph, there exists an edge labeling $\gamma'$ that saturates $V_1$ with respect to $w'$. Observe that by decreasing the weights of $\gamma'$ slightly, we easily obtain an edge labeling $\gamma$ that saturates $V_1$ with respect to the original weighting $w$.

Finally, consider the most general case: condition (19) holds and $w(v) \in \mathbb{R}$ for all $v \in V$. Note that the problem of finding an edge labeling that saturates $V_1$ may be rephrased as follows. Let $b_1 \in \mathbb{R}^{|V_1|}$ be the vector of weights $(w(s) \colon s \in V_1)$. Then for an appropriate choice of the matrix $A_1 \in \{0,1\}^{|V_1| \times |E|}$, the conditions

$$\sum_{t \in N(s)} \gamma_{st} = w(s), \qquad \forall s \in V_1,$$

may be expressed as a system of linear equations,

$$A_1 \gamma = b_1. \tag{21}$$

Similarly, letting $b_2 = (w(t) \colon t \in V_2) \in \mathbb{R}^{|V_2|}$, the conditions

$$\sum_{s \in N(t)} \gamma_{st} \leq w(t), \qquad \forall t \in V_2,$$

may be expressed in the form

$$A_2 \gamma \leq b_2, \tag{22}$$

where $A_2 \in \{0,1\}^{|V_2| \times |E|}$. A saturating edge labeling exists if and only if there exists $\gamma \in \mathbb{R}^{|E|}_{\geq 0}$ that simultaneously satisfies conditions (21) and (22). Now consider a sequence of weight vectors $\{b_1^n\}_{n \geq 1}$, such that $b_1^n \to b_1$ and the convergence is from below and strictly monotone for each component. Let $w^n = (b_1^n, b_2)$ denote the full sequence of weights. Then

$$w^n(U) < w(U) \leq w(N(U)) = w^n(N(U)), \qquad \forall U \subseteq V.$$

It follows by the result of the previous paragraph that there exists an edge labeling $\gamma^n \in \mathbb{R}^{|E|}_{\geq 0}$ such that

$$A_1 \gamma^n = b_1^n, \qquad \text{and} \qquad \gamma^n \in D := \left\{ \gamma \in \mathbb{R}^{|E|}_{\geq 0} \colon A_2 \gamma \leq b_2 \right\}.$$

Clearly, $D$ is a closed set; furthermore, it is easy to see that the constraint $A_2 \gamma \leq b_2$ implies that each component of $\gamma$ is bounded from above, since $A_2$ contains only nonnegative entries. It follows that the sequence $\{\gamma_n\}_{n \geq 1}$ has a limit point $\gamma^* \in D$. By continuity of the linear map $A_1$, we must have

$$A_1 \gamma^* = \lim_{n \to \infty} A_1 \gamma_n = \lim_{n \to \infty} b_1^n = b_1.$$

Hence, $\gamma^*$ is a valid edge labeling that saturates $V_1$.

$\square$

### A.3  Proof of Corollary 1

By Theorem 1, $H(\tau; \rho)$ is strictly concave provided condition (10) holds. Note that

$$\mathcal{F}(\alpha) = \{\alpha\}, \qquad \forall \alpha \in F,$$

whereas

$$\mathcal{F}(s) = \{s\} \cup N(s), \qquad \forall s \in V.$$

Condition (10) applied to the set $S = \{\alpha\}$ gives the inequality

$$\rho_\alpha \geq 0, \qquad \forall \alpha \in F. \tag{23}$$

For a subset $U \subseteq V$, we can write

$$\mathcal{F}(U) = \bigcup_{s \in U} \mathcal{F}(s) = U \cup N(U) = U \cup \{\alpha \in F \colon \alpha \cap U \neq \emptyset\},$$

so (10) translates into

$$\sum_{s \in U} \rho_s + \sum_{\alpha \in F \colon \alpha \cap U \neq \emptyset} \rho_\alpha \geq 0, \qquad \forall U \subseteq V, \tag{24}$$

which is condition (12). It is easy to see that conditions (23) and (24) together also imply the validity of condition (10) for any other set of regions $S \subseteq R$.

## B  Proofs for Section 3.2

### B.1  Proof of Theorem 2

Our result relies on the property that if the Bethe entropy $H(\tau; \rho)$ is concave over $\Delta_R^K$, then $H(\tau; \rho)$ is also concave over any subset $\Delta' \subseteq \Delta_R^K$. In particular, it is sufficient to assume that $\mathcal{X}$ is binary, say $\mathcal{X} = \{-1, +1\}$; the general multinomial case $|\mathcal{X}| > 2$ follows by restricting the distribution of $X_s$ to be supported on only two points.

The first lemma shows that $\rho_\alpha \geq 0$ for all $\alpha \in F$. The proof is contained in Appendix B.2.

**Lemma 2.** *If the Bethe entropy $H(\tau; \rho)$ is concave over $\Delta_R^K$, then $\rho_\alpha \geq 0$ for all $\alpha \in F$.*

To establish the necessity of condition (12), consider a nonempty subset $U \subseteq V$ and the corresponding sub-region graph $R_U = U \cup F_U$, where $F_U = \{\alpha \cap U : \alpha \in F, \alpha \cap U \neq \emptyset\}$. From the original weights $\rho \in \mathbb{R}^{|V|+|F|}$, construct the sub-region weights $\rho^U \in \mathbb{R}^{|U|+|F_U|}$ given by

$$\rho_s^U = \rho_s, \quad \forall s \in U, \qquad \text{and} \qquad \rho_{\alpha \cap U}^U = \rho_\alpha, \quad \forall \alpha \cap U \in F_U.$$

For simplicity, we consider $R_U$ to be a multiset by remembering which factor $\alpha \in F$ each $\beta = \alpha \cap U \in F_U$ comes from; we can equivalently work with $R_U$ as a set by defining the weights $\rho^U$ to be the sum over the pre-images of the factors in $R_U$. Consider the set of locally consistent $R_U$-pseudomarginals $\Delta_{R_U}^K$. Define a map that sends $\tilde{\tau} \in \Delta_{R_U}^K$ to $\tau \in \Delta_R^K$ defined by

$$\tau_s(x_s) = \begin{cases} \tilde{\tau}_s(x_s) & \text{if } s \in U, \\ \frac{1}{2} & \text{otherwise,} \end{cases}$$

$$\tau_\alpha(x_\alpha) = \begin{cases} \tilde{\tau}_{\alpha \cap U}(x_{\alpha \cap U}) \cdot \prod_{s \in \alpha \setminus U} \tau_s(x_s) & \text{if } \alpha \cap U \neq \emptyset \ (\text{so } \alpha \cap U \in F_U), \\ \prod_{s \in \alpha} \tau_s(x_s) & \text{otherwise.} \end{cases}$$

Let $\Delta_U$ denote the image of $\Delta_{R_U}^K$ under the mapping above, and note that $\Delta_U \subseteq \Delta_R^K$. Therefore, $H(\tau; \rho)$ is concave over $\Delta_U$. Now let $\tau \in \Delta_U$ and let $\tilde{\tau} \in \Delta_{R_U}^K$ be a pre-image of $\tau$. With this construction, we have the following lemma, proved in Appendix B.3:

**Lemma 3.** *The entropy $H(\tau; \rho)$ differs from $H_U(\tilde{\tau}; \rho^U)$ by a constant, where $H_U(\tilde{\tau}; \rho^U)$ is the Bethe entropy defined over the sub-region graph $R_U$.*

Finally, we have a lemma showing that we can extract condition (12) for $U = V$. The proof is provided in Appendix B.4.

**Lemma 4.** *If the Bethe entropy $H(\tau; \rho)$ is concave over $\Delta_R^K$, then $\sum_{s \in V} \rho_s + \sum_{\alpha \in F} \rho_\alpha \geq 0$.*

By Lemma 3, the concavity of $H(\tau; \rho)$ over $\Delta_U$ implies the concavity of $H_U(\tilde{\tau}; \rho^U)$ over $\Delta_{R_U}^K$. Then by Lemma 4 applied to $R_U$, we have

$$\sum_{s \in U} \rho_s + \sum_{\alpha \in F : \alpha \cap U \neq \emptyset} \rho_\alpha = \sum_{s \in U} \rho_s^U + \sum_{\beta \in F_U} \rho_\beta^U \geq 0,$$

finishing the proof.

## B.2 Proof of Lemma 2

Fix $\alpha \in F$, and let $\Delta_\alpha$ be the set of pseudomarginals $\tau \in \Delta_R^K$ with the property that for all $s \in V$ and $\beta \in F \setminus \{\alpha\}$, $\tau_s$ and $\tau_\beta$ are uniform distributions over $X_s$ and $X_\beta$, respectively, while $\tau_\alpha$ is an arbitrary distribution on $X_\alpha$ with uniform single-node marginals. Then $H(\tau; \rho)$ is concave over $\Delta_\alpha$. On the other hand, note that for $\tau \in \Delta_\alpha$, $H_s(\tau_s) = \log 2$ and $H_\beta(\tau_\beta) = |\beta| \log 2$ are constants for $s \in V$ and $\beta \in F \setminus \{\alpha\}$, so we can write

$$H(\tau; \rho) = \rho_\alpha H_\alpha(\tau_\alpha) + \text{constant}.$$

Since $H_\alpha(\tau_\alpha)$ is concave in $\tau_\alpha$, this implies $\rho_\alpha \geq 0$, as claimed.

## B.3 Proof of Lemma 3

By construction, for $s \in V \setminus U$, we have $H_s(\tau_s) = \log 2$; and for $\alpha \in F$ with $\alpha \cap U = \emptyset$, we have $H_\alpha(\tau_\alpha) = |\alpha| \log 2$. Moreover, for $\alpha \in F$ with $\alpha \cap U \neq \emptyset$, we have

$$H_\alpha(\tau_\alpha) = H_{\alpha \cap U}(\tilde{\tau}_{\alpha \cap U}) + \sum_{s \in \alpha \setminus U} H_s(\tau_s) = H_{\alpha \cap U}(\tilde{\tau}_{\alpha \cap U}) + |\alpha \setminus U| \log 2.$$

Therefore, for $\tau \in \Delta_U$, we can write

$$
\begin{aligned}
H(\tau; \rho) &= \sum_{s \in V} \rho_s H_s(\tau_s) + \sum_{\alpha \in F} \rho_\alpha H_\alpha(\tau_\alpha) \\
&= \sum_{s \in U} \rho_s H_s(\tilde{\tau}_s) + \Big( \sum_{s \in V \setminus U} \rho_s \Big) \log 2 \\
&\quad + \sum_{\alpha \in F : \alpha \cap U \neq \emptyset} \rho_\alpha \Big( H_{\alpha \cap U}(\tilde{\tau}_{\alpha \cap U}) + |\alpha \setminus U| \log 2 \Big) + \sum_{\alpha \in F : \alpha \cap U = \emptyset} \rho_\alpha |\alpha| \log 2 \\
&= \sum_{s \in U} \rho_s^U H_s(\tilde{\tau}_s) + \sum_{\beta \in F_U} \rho_\beta^U H_\beta(\tilde{\tau}_\beta) + \text{constant} \\
&= H_U(\tilde{\tau}; \rho^U) + \text{constant},
\end{aligned}
$$

as wanted.

## B.4   Proof of Lemma 4

Given $m_o, m_e \in \mathbb{R}$, we define a pseudomarginal $\tau = (\tau_s, \tau_\alpha)$ by[1]

$$
\tau_s(x_s) = \frac{1 + x_s m_o}{2}, \qquad \forall s \in V, \ x_s \in X = \{-1, +1\},
$$

and for $\alpha \in F$ with $|\alpha| = k$,

$$
\tau_\alpha(x_\alpha) = \begin{cases} 2^{-k} \big( 1 + 2^{k-1} m_o + (2^{k-1} - 1) m_e \big) & \text{if } x_\alpha = (1, \ldots, 1), \\ 2^{-k} \big( 1 - 2^{k-1} m_o + (2^{k-1} - 1) m_e \big) & \text{if } x_\alpha = (-1, \ldots, -1), \\ 2^{-k} (1 - m_e) & \text{otherwise.} \end{cases}
$$

It is easy to see that $\sum_{x_s} \tau_s(x_s) = \sum_{x_\alpha} \tau_\alpha(x_\alpha) = 1$, and that $\tau_s$ is the single-node marginal of $\tau_\alpha$. Thus, for $\tau$ to lie in $\Delta_R^K$, we only need to ensure that $\tau_s(x_s) \geq 0$ and $\tau_\alpha(x_\alpha) \geq 0$, or equivalently,

$$
-1 \leq m_o \leq 1, \qquad \frac{1 + 2^{k-1} |m_o|}{2^{k-1} - 1} \leq m_e \leq 1, \qquad \forall 2 \leq k \leq K,
$$

where $K = \max\{|\alpha| : \alpha \in F\}$. Let $M$ denote the set of $(m_o, m_e)$ satisfying the constraints above, and let $\Delta_M$ denote the set of pseudomarginals $\tau[m_o, m_e] \in \Delta_R^K$ given by the construction above for each $(m_o, m_e) \in M$.

Observe that the function $(m_o, m_e) \mapsto \tau[m_o, m_e]$ is additive for convex combinations; i.e., for $(m_o^{(1)}, m_e^{(1)}), \ldots, (m_o^{(j)}, m_e^{(j)}) \in M$ and $\lambda_1, \ldots, \lambda_j \geq 0$ with $\lambda_1 + \cdots + \lambda_j = 1$, we have

$$
\sum_{i=1}^{j} \lambda_i \tau[m_o^{(i)}, m_e^{(i)}] = \tau\Big[ \sum_{i=1}^{j} \lambda_i m_o^{(i)}, \ \sum_{i=1}^{j} \lambda_i m_e^{(i)} \Big].
$$

Since $M$ is convex, this shows that $\Delta_M$ is a convex subset of $\Delta_R^K$. Therefore, $H(\tau; \rho)$ is concave over $\Delta_M$, and the additivity property above implies that the function

$$
\zeta(m_o, m_e) := H(\tau[m_o, m_e]; \rho)
$$

is concave over $M$. We now compute the Hessian of $\zeta$ and show how it relates to the required quantity that we want to prove is nonnegative.

Fix $(m_o, m_e) \in M$, and note that $\tau \equiv \tau[m_o, m_e]$ has the property that $\tau_\alpha = \tau_\beta$ whenever $|\alpha| = |\beta|$. Therefore, we can collect the terms in $H(\tau; \rho)$ based on the cardinality of $\alpha \in V \cup F$. The single-node entropy is, as a function of $m_o$,

$$
\zeta_1(m_o) := H_s(\tau_s) = -\eta\left(\frac{1 + m_o}{2}\right) - \eta\left(\frac{1 - m_o}{2}\right),
$$

$$
\mathbb{E}_{\tau_\alpha}\Big[ \prod_{s \in \beta} X_s \Big] = m_o \ \text{ if } |\beta| \text{ is odd} \quad \text{and} \quad \mathbb{E}_{\tau_\alpha}\Big[ \prod_{s \in \beta} X_s \Big] = m_e \ \text{ if } |\beta| \text{ is even,}
$$

for all $\alpha \in V \cup F$ and $\emptyset \neq \beta \subseteq \alpha$.

where $\eta(t) := t \log t$. For $\alpha \in F$ with $|\alpha| = k \geq 2$, the entropy corresponding to $\tau_\alpha$ is

$$\zeta_k(m_o, m_e) := H_\alpha(\tau_\alpha) = -\eta\left(\frac{1 + 2^{k-1}m_o + (2^{k-1} - 1)m_e}{2^k}\right) - \eta\left(\frac{1 - 2^{k-1}m_o + (2^{k-1} - 1)m_e}{2^k}\right)$$
$$- (2^k - 2)\,\eta\left(\frac{1 - m_e}{2^k}\right).$$

The Bethe entropy can then be written as

$$\zeta(m_o, m_e) = H(\tau; \rho) = c_1\zeta_1(m_o) + \sum_{k=2}^{K} c_k\zeta_k(m_o, m_e),$$

where $c_1 = \sum_{s \in V} \rho_s$ and $c_k = \sum_{\alpha \in F:\ |\alpha| = k} \rho_\alpha$ for $k \geq 2$.

Let us compute the Hessian matrix of $\zeta(m_o, m_e)$ along the axis $m_o = 0$. The function $\zeta_1$ has second derivative $\zeta_1''(m_o) = -1/(1 - m_o^2)$, so at $m_o = 0$, the contribution of $\zeta_1$ to the Hessian of $\zeta$ is

$$\nabla^2\zeta_1(0, m_e) = \begin{pmatrix} -1 & 0 \\ 0 & 0 \end{pmatrix}.$$

For $k \geq 2$, the first partial derivatives of $\zeta_k$ are

$$\frac{\partial\zeta_k}{\partial m_o} = -\frac{1}{2}\left\{\log\left(1 + 2^{k-1}m_o + (2^{k-1} - 1)m_e\right) - \log\left(1 - 2^{k-1}m_o + (2^{k-1} - 1)m_e\right)\right\},$$

$$\frac{\partial\zeta_k}{\partial m_e} = -\frac{(2^{k-1} - 1)}{2^k}\left\{\log\left(1 + 2^{k-1}m_o + (2^{k-1} - 1)m_e\right) + \log\left(1 - 2^{k-1}m_o + (2^{k-1} - 1)m_e\right)\right.$$
$$\left. - 2\log\left(1 - m_e\right)\right\}.$$

The Hessian $\nabla^2\zeta_k$ at $m_o = 0$ is then given by

$$\nabla^2\zeta_k(0, m_e) = \begin{pmatrix} -\dfrac{2^{k-1}}{1 + (2^{k-1} - 1)m_e} & 0 \\ 0 & -\dfrac{2^{k-1} - 1}{(1 + (2^{k-1} - 1)m_e)(1 - m_e)} \end{pmatrix}.$$

Therefore, the Hessian of $\zeta$ at $m_o = 0$ is the diagonal matrix

$$\nabla^2\zeta(0, m_e) = \begin{pmatrix} -c_1 - \displaystyle\sum_{k=2}^{K}\dfrac{2^{k-1}c_k}{1 + (2^{k-1} - 1)m_e} & 0 \\ 0 & -\displaystyle\sum_{k=2}^{K}\dfrac{(2^{k-1} - 1)c_k}{(1 + (2^{k-1} - 1)m_e)(1 - m_e)} \end{pmatrix}.$$

In particular, the eigenvalues of $\nabla^2\zeta(0, m_e)$ are its diagonal entries. Taking $m_e \to 1$, we see that the eigenvalue corresponding to the first diagonal entry satisfies

$$\lim_{m_e \to 1} \lambda_1(m_e) = \lim_{m_e \to 1}\left\{-c_1 - \sum_{k=2}^{K}\frac{2^{k-1}c_k}{1 + (2^{k-1} - 1)m_e}\right\} = -\sum_{k=1}^{K} c_k.$$

Since $(0, m_e) \in M$ as $m_e \to 1$ and $\zeta(m_o, m_e)$ is concave over $M$, we see that the eigenvalue above is nonpositive, which implies

$$\sum_{s \in V} \rho_s + \sum_{\alpha \in F} \rho_\alpha = \sum_{k=1}^{K} c_k \geq 0,$$

as desired.

# C  Proofs for Section 3.3

## C.1  Proof of Theorem 3

We first show that $\mathrm{conv}(\mathbb{F}) \subseteq \mathbb{C}$ in the general Bethe case. Since $\mathbb{C}$ is convex, it suffices to show that $\mathbb{F} \subseteq \mathbb{C}$, so consider $1_{F'} \in \mathbb{F}$. We need to show that inequality (15) holds for $\rho = 1_{F'}$.

Let $W_1, \ldots, W_m$ denote the connected components of $F' \cup N(F')$ in $G$. Consider an arbitrary $U \subseteq V$, and define $U_i := W_i \cap U$ for $1 \leq i \leq m$, and $U_0 := U \backslash \{U_1, \ldots, U_m\}$. Then each $W_i$ has at most one cycle. Furthermore, we may write

$$\sum_{\substack{\alpha \in F: \\ \alpha \cap U \neq \emptyset}} (|\alpha \cap U| - 1)\rho_\alpha = \sum_{\substack{\alpha \in F': \\ \alpha \cap U \neq \emptyset}} (|\alpha \cap U| - 1) = \sum_{i=1}^{m} \left\{ \sum_{\substack{\alpha \in W_i: \\ \alpha \cap U_i \neq \emptyset}} (|\alpha \cap U_i| - 1) \right\}. \qquad (25)$$

We claim that

$$\sum_{\alpha \in W_i:\ \alpha \cap U_i \neq \emptyset} (|\alpha \cap U_i| - 1) \leq |U_i|, \qquad \forall 1 \leq i \leq m. \qquad (26)$$

Indeed, consider the induced subgraph $W_i'$ of $W_i$ with vertex set $V_i := U_i \cup \{\alpha \in W_i : \alpha \cap U_i \neq \emptyset\}$. Since $W_i$ has at most one cycle, $W_i'$ has at most one cycle, as well. Furthermore, the number of edges of $W_i'$ is given by

$$|E(W_i')| = \sum_{\alpha \in W_i:\ \alpha \cap U_i \neq \emptyset} |\alpha \cap U_i|,$$

and the number of vertices is $|V_i| = |U_i| + |\{\alpha \in W_i : \alpha \cap U_i \neq \emptyset\}|$.

We have the following simple lemma:

**Lemma 5.** *A connected graph $G$ has at most one cycle if and only if*

$$|E(U)| \leq |U|, \qquad \forall U \subseteq V.$$

*Proof.* First suppose $G$ has at most one cycle. For any subset $U \subseteq V$, the induced subgraph $H$ clearly also contains at most one cycle. Hence, we may remove at most one edge to obtain a graph $H'$ which is a forest. Then

$$|E(H')| \leq |V(H')| - 1 = |U| - 1. \qquad (27)$$

Furthermore, $|E(U)| \leq |E(H')| + 1$. It follows that $|E(U)| \leq |U|$.

Conversely, if $G$ is a connected graph with more than one cycle, we may pick $U$ to be the union of vertices in the two cycles, along with a path connecting the two cycles (in case the cycles are disconnected). It is easy to check that condition (27) is violated in this case. $\square$

Applying Lemma 5 to the graph $W_i'$ and rearranging then yields inequality (26). Combining with equation (25) then yields

$$\sum_{\alpha \in F:\ \alpha \cap U \neq \emptyset} (|\alpha \cap U| - 1)\rho_\alpha \leq \sum_{i=1}^{m} |U_i| = |U| - |U_0| \leq |U|,$$

proving the condition (15).

We now specialize to the case where $|\alpha| = 2$ for all $\alpha \in F$. Note that in this case, we may identify the region graph with an ordinary graph $\overline{G} = (V, E)$, where the edge set $E$ is given by $F$. It is easy to check that $1_{F'} \in \mathbb{F}$ if and only if the subgraph of $\overline{G}$ with edge set $F'$ is a single-cycle forest. In the following argument, we abuse notation and refer to $\overline{G}$ as $G$.

Recall that a *rational polyhedron* is a set of the form $\{x \in \mathbb{R}^p : Ax \leq b\}$, such that $A$ and $b$ have rational entries. Clearly, $\mathbb{C}$ is a rational polyhedron. Furthermore, a polyhedron is *integral* if all vertices are elements of the integer lattice $\mathbb{Z}^p$. The following result is standard in integer programming:

**Lemma 6.** *[Theorem 5.12, [10]] Let $P$ be a rational polyhedron. Then $P$ is integral if and only if* $\max\{c^T x : x \in P\}$ *is attained by an integral vector for each $c$ for which the maximum is finite.*

We have already established that $1_{F'} \in \mathbb{C}$ for all $1_{F'} \in \mathbb{F}$. Furthermore, any lattice point in $\mathbb{C}$ is of the form $1_H$, where $H \subseteq E$. By Lemma 5, each connected component of $H$ must contain at most one cycle, implying that $H$ is a single-cycle forest. Hence, $1_H \in \mathbb{F}$, as well. We then combine Lemma 6 with the following proposition to obtain the desired result.

**Proposition 2.** *Let $G = (V, E)$ be a graph. For any set of weights $c = (c_{st}) \in \mathbb{R}^{|E|}$, the LP*

$$\max \sum_{(s,t)\in E} c_{st} x_{st} \tag{28}$$

$$s.t. \sum_{(s,t)\in E(U)} x_{st} \leq |U|, \qquad \forall U \subseteq V, \tag{29}$$

$$0 \leq x_{st} \leq 1, \qquad \forall (s,t) \in E,$$

*attains its maximum value at an integral vector $x^*$.*

*Proof.* We first argue that it suffices to consider rational weights $c \in \mathbb{Q}^{|E|}$. Let $X$ denote the feasible set of the LP, and let $F(c) = \max_{x\in X} c^\top x$ denote the maximum value of the LP. Note that $F(c)$ is continuous in $c$.

Suppose the claim in the proposition holds for $c \in \mathbb{Q}^{|E|}$. Given $c \in \mathbb{R}^{|E|}$, let $x^* \in \arg\max_{x\in X} c^T x$. Let $(c^{(n)})_{n\geq 1}$ be a sequence of weights in $\mathbb{Q}^{|E|}$ converging to $c$ elementwise as $n \to \infty$. Given $\epsilon > 0$, choose $n$ sufficiently large such that $\|c^{(n)} - c\|_1 < \epsilon$ and $|F(c) - F(c^{(n)})| < \epsilon$. Applying our hypothesis, we know there exists an integral vector $z^* \in X$ such that $F(c^{(n)}) = (c^{(n)})^\top z^*$. Then

$$|F(c) - c^\top z^*| \leq |F(c) - F(c^{(n)})| + |(c^{(n)} - c)^\top z^*| \leq \epsilon + \|c^{(n)} - c\|_1 \|z^*\|_\infty \leq 2\epsilon.$$

Thus, we can find an integral vector $z^* \in X$ that achieves the objective function that is within $2\epsilon$ from the optimal value. Since $\epsilon > 0$ is arbitrary, we conclude by continuity that we may find an integral vector in $X$ arbitrarily close to $x^*$. This implies that $x^*$ is an integral vector.

It now remains to prove the claim in the proposition for $c \in \mathbb{Q}^{|E|}$. If $c_{st} < 0$ for some $(s,t) \in E$, then any optimal solution $x^*$ will have $x_{st}^* = 0$. If $c_{st} = 0$, then we can set $x_{st}^* = 0$ without changing the objective value. Thus, we can assume $c_{st} > 0$ for all $(s,t) \in E$. By scaling the weights, we can further assume that $c_{st} \in \{1, \ldots, K\}$ for all $(s,t) \in E$, for some $K \in \mathbb{N}$.

We first upper-bound the objective function. For $1 \leq i \leq K$, let $E_i = \{(s,t) \in E : c_{st} \geq i\}$ denote the set of edges with weights at least $i$, and let $V_i$ denote the set of vertices in $E_i$. By construction, we have

$$V = V_1 \supset \cdots \supset V_K, \qquad \text{and} \qquad E = E_1 \supset \cdots \supset E_K.$$

Suppose the subgraph $G_i = (V_i, E_i)$ is decomposed into connected components

$$G_i = T_{i1} \cup \cdots T_{i\alpha_i} \cup H_{i1} \cup \cdots \cup H_{i\beta_i}, \tag{30}$$

where each $T_{ij} = (V(T_{ij}), E(T_{ij}))$ is a tree and each $H_{i\ell} = (V(H_{i\ell}), E(H_{i\ell}))$ is a connected graph with at least one loop. Thus, we have the disjoint partitions

$$V_i = \bigcup_{j=1}^{\alpha_i} V(T_{ij}) \cup \bigcup_{\ell=1}^{\beta_i} V(H_{i\ell}), \qquad \text{and} \qquad E_i = \bigcup_{j=1}^{\alpha_i} E(T_{ij}) \cup \bigcup_{\ell=1}^{\beta_i} E(H_{i\ell}).$$

Then we can write the objective function of the LP as

$$\sum_{(s,t)\in E} c_{st} x_{st} = \sum_{i=1}^{K} \sum_{(s,t)\in E_i} x_{st} = \sum_{i=1}^{K} \left( \sum_{j=1}^{\alpha_i} \sum_{(s,t)\in E(T_{ij})} x_{st} + \sum_{\ell=1}^{\beta_i} \sum_{(s,t)\in E(H_{i\ell})} x_{st} \right). \tag{31}$$

For $i = 1, \ldots, K$ and $j = 1, \ldots, \alpha_i$, since $T_{ij}$ is a tree, we have

$$\sum_{(s,t)\in E(T_{ij})} x_{st} \leq |E(T_{ij})| = |V(T_{ij})| - 1, \qquad \forall x \in X. \tag{32}$$

For $\ell = 1, \ldots, \beta_i$, note that the set $E(H_{i\ell})$ of edges in $H_{i\ell}$ is contained within the set $E(V(H_{i\ell}))$ of edges in the subgraph of $G$ induced by $V(H_{i\ell})$. Thus, by inequality (29), we have

$$\sum_{(s,t)\in E(H_{i\ell})} x_{st} \leq \sum_{(s,t)\in E(V(H_{i\ell}))} x_{st} \leq |V(H_{i\ell})|. \tag{33}$$

Plugging in the bounds (32) and (33) to inequality (31), we arrive at the upper bound

$$\sum_{(s,t) \in E} c_{st} x_{st} \leq \sum_{i=1}^{K} \left( \sum_{j=1}^{\alpha_i} \{|V(T_{ij})| - 1\} + \sum_{\ell=1}^{\beta_i} |V(H_{i\ell})| \right) = \sum_{i=1}^{K} (|V_i| - \alpha_i). \qquad (34)$$

We now prove the claim in the proposition by explicitly constructing an integral vector $x^*$ that achieves the upper bound (34). Since $x^* \in \{0,1\}^{|E|}$, it is the indicator vector of a subset $E^* \subseteq E$.

Our approach is to construct, for each $1 \leq i \leq K$, a spanning single-cycle forest $F_i = (V_i, C_i)$ of $G_i = (V_i, E_i)$ with the following properties:

1. The restriction of $F_i$ to $V_{i+1} \subseteq V_i$ is equal to $F_{i+1} = (V_{i+1}, C_{i+1})$, or equivalently, $C_i \cap E_{i+1} = C_{i+1}$. By induction, this implies $C_1 \cap E_i = C_i$, for $1 \leq i \leq K$.

2. For $1 \leq i \leq K$, we have $|C_i| = |V_i| - \alpha_i$.

Suppose we can construct such $F_i$'s. Setting $E^* = C_1$, we see that this construction yields a vector $x^* = 1_{E^*}$ satisfying

$$\sum_{(s,t) \in E} c_{st} x_{st}^* = \sum_{i=1}^{K} \sum_{(s,t) \in E_i} x_{st}^* = \sum_{i=1}^{K} \sum_{(s,t) \in E_i} \mathbf{1}\{(s,t) \in C_1\}$$

$$= \sum_{i=1}^{K} |C_1 \cap E_i| = \sum_{i=1}^{K} |C_i| = \sum_{i=1}^{K} (|V_i| - \alpha_i),$$

so $x^*$ achieves the bound (34), as desired.

It now remains to construct the $F_i$'s. We start by taking $F_K$ to be a spanning single-cycle forest of $G_K$. Specifically, for each connected component $H$ of $G_K$, we do the following: If $H$ is a tree, we take $H$ to be in $F_K$. If $H$ contains at least one loop, then we take an arbitrary spanning single-cycle subgraph (i.e., a spanning tree with an additional edge to form one cycle) of $H$ to be in $F_K$. Then $F_K = (V_K, C_K)$ satisfies $|C_K| = |V_K| - \alpha_K$, since there are $\alpha_K$ trees among the connected components of $G_K$.

Suppose that for some $1 \leq i \leq K - 1$, we have constructed a spanning single-cycle forest $F_{i+1}$ satisfying the desired properties. Now consider $G_i = (V_i, E_i)$, and construct $F_i = (V_i, C_i)$ as follows: Consider each connected component of $G_i$ in the decomposition (30).

(a) For each tree $T_{ij} = (V(T_{ij}), E(T_{ij}))$, for all $1 \leq j \leq \alpha_i$, take $T_{ij}$ to be in $F_i$. This component of $F_i$ is clearly consistent with $F_{i+1}$, and the contribution to the total number of edges $|C_i|$ is

$$\sum_{j=1}^{\alpha_i} |E(T_{ij})| = \sum_{j=1}^{\alpha_i} (|V(T_{ij})| - 1) = \sum_{j=1}^{\alpha_i} |V(T_{ij})| - \alpha_i.$$

(b) Consider $H_{i\ell} = (V(H_{i\ell}), E(H_{i\ell}))$, for some $1 \leq \ell \leq \beta_i$, so $H_{i\ell}$ has at least one loop. There may be several connected components of $F_{i+1}$ in $H_{i\ell}$; suppose there are $\gamma_{i\ell}$ trees and $\delta_{i\ell}$ single-cycle graphs from $F_{i+1}$ in $H_{i\ell}$. From each of the $\delta_{i\ell}$ single-cycle graphs, remove one edge to reduce it to a tree, and complete the $\gamma_{i\ell} + \delta_{i\ell}$ trees into a spanning tree of $H_{i\ell}$. Add the $\delta_{i\ell}$ edges back, so the spanning tree now has $\delta_{i\ell}$ cycles. Remove $\delta_{i\ell} - 1$ edges to break this graph into $\delta_{i\ell}$ connected components, such that each of the original $\delta_{i\ell}$ single-cycle graphs is in a separate connected components, and the last connected component is a tree. Set this new graph to be in $F_i$. It is clear by construction that this component of $F_i$ is consistent with $F_{i+1}$ since we keep all the edges from $F_{i+1}$. Moreover, its contribution to the total number of edges $C_i$ is precisely

$$\sum_{\ell=1}^{\beta_i} (\{|V(H_{i\ell})| - 1\} + \delta_{i\ell} - \{\delta_{i\ell} - 1\}) = \sum_{\ell=1}^{\beta_i} |V(H_{i\ell})|.$$

Combining the two cases above, for each $1 \leq i \leq K$ we have constructed a spanning single-cycle forest $F_i$ that is consistent with $F_{i+1}$ and satisfies $|C_i| = \sum_{j=1}^{\alpha_i} |V(T_{ij})| - \alpha_i + \sum_{\ell=1}^{\beta_i} |V(H_{i\ell})| = |V_i| - \alpha_i$, as desired. This completes the proof of the proposition.

$\square$

## C.2 Details for Example 1

It is easy to check that $\mathbb{F} = \{0,1\}^3 \backslash (1,1,1)$. Hence, $(1, \frac{1}{2}, 1) \notin \text{conv}(\mathbb{F})$. By enumerating the inequalities defining the boundary of $\mathbb{C}$ for different values of $U \subseteq V$, one may check that the only inequalities that are not trivially satisfied by $\rho \in [0,1]^3$ are

$$\rho_1 + 2\rho_2 + \rho_3 \leq 3,$$
$$2\rho_1 + 2\rho_2 + \rho_3 \leq 4,$$
$$\rho_1 + 2\rho_2 + 2\rho_3 \leq 4,$$
$$2\rho_1 + 2\rho_2 + 2\rho_3 \leq 5.$$

The first inequality together with the condition $\rho \in [0,1]^3$ implies the remaining three inequalities, so

$$\mathbb{C} = \left\{ \rho \in [0,1]^3 \colon \rho_1 + 2\rho_2 + \rho_3 \leq 3 \right\}.$$

Clearly, $(1, \frac{1}{2}, 1) \in \mathbb{C}$.

## C.3 Proof of Proposition 1

The first condition implies $F \notin \mathfrak{F}$. In particular, $F^* \neq F$ and we can find $\alpha^* \in F \setminus F^*$. Since $F^*$ is maximal, $\tilde{F} = F^* \cup \{\alpha^*\} \notin \mathfrak{F}$. This means $1_{F^*} \in \mathbb{F}$ but $1_{\tilde{F}} = 1_{F^*} + 1_{\{\alpha^*\}} \notin \mathbb{F}$. Define

$$\rho = 1_{F^*} + \epsilon 1_{\{\alpha^*\}}, \qquad \text{with} \qquad \epsilon = \frac{1}{|\alpha^*| - 1} \in (0,1).$$

We claim that $\rho \in \mathbb{C}$, which will give us the desired conclusion since $\rho \notin \text{conv}(\mathbb{F})$.

To show $\rho \in \mathbb{C}$, since we already know that $1_{F^*} \in \mathbb{F} \subseteq \mathbb{C}$, we only need to verify inequality (15) for $U \subseteq V$ with $U \cap \alpha^* \neq \emptyset$. Given such a subset $U$, note that since $F^* \cup N(F^*)$ is a forest, the subgraph induced by the nodes $U \cup \{\alpha \in F^* \colon \alpha \cap U \neq \emptyset\}$ is also a forest, so

$$\sum_{\alpha \in F^* \colon \alpha \cap U \neq \emptyset} (|\alpha \cap U| - 1) \leq |U| - 1.$$

Therefore,

$$\sum_{\alpha \in F \colon \alpha \cap U \neq \emptyset} (|\alpha \cap U| - 1)\rho_\alpha = \sum_{\alpha \in F^* \colon \alpha \cap U \neq \emptyset} (|\alpha \cap U| - 1) + \frac{|\alpha^* \cap U| - 1}{|\alpha^*| - 1} \leq |U| - 1 + 1 = |U|,$$

verifying condition (15), as desired.

# D   Proof of Theorem 4

For $r \in R$ and $s \in \mathcal{P}(r)$, let $\lambda_{sr}(x_r)$ be a Lagrange multiplier associated with the consistency constraint $\sum_{x_{s\backslash r}} \tau_s(x_r, x_{s\backslash r}) = \tau_r(x_r)$. We enforce the nonnegativity constraint $\tau_r(x_r) \geq 0$ and normalization constraint $\sum_{x_r} \tau_r(x_r) = 1$ explicitly. Then the Lagrangian associated with the optimization problem (8) is

$$\mathcal{L}_{\theta,\rho}(\tau; \lambda) = \sum_{r \in R} \sum_{x_r} \tau_r(x_r)\theta_r(x_r) - \sum_{r \in R} \rho_r \sum_{x_r} \tau_r(x_r) \log \tau_r(x_r)$$

$$+ \sum_{r \in R} \sum_{t \in \mathcal{C}(r)} \sum_{x_t} \lambda_{rt}(x_t) \left( \tau_t(x_t) - \sum_{x_{r\backslash t}} \tau_r(x_t, x_{r\backslash t}) \right). \tag{35}$$

Setting the partial derivatives of $\mathcal{L}_{\theta,\rho}$ with respect to the Lagrange multipliers equal to zero recovers the consistency constraints. Taking the derivative of $\mathcal{L}_{\theta,\rho}$ with respect to $\tau_r(x_r)$ and setting it equal to zero yields

$$\log \tau_r(x_r) = C + \frac{\theta_r(x_r)}{\rho_r} + \sum_{s \in \mathcal{P}(r)} \frac{\lambda_{sr}(x_r)}{\rho_r} - \sum_{t \in \mathcal{C}(r)} \frac{\lambda_{rt}(x_t)}{\rho_r},$$

where $C$ is a constant that enforces the normalization condition $\sum_{x_r} \tau_r(x_r) = 1$. Defining the messages by

$$\log M_{sr}(x_r) = \frac{\lambda_{sr}(x_r)}{\rho_s},$$

we can write the equation above as

$$\tau_r(x_r) \propto \exp\left(\frac{\theta_r(x_r)}{\rho_r}\right) \frac{\prod_{s \in \mathcal{P}(r)} M_{sr}(x_r)^{\rho_s/\rho_r}}{\prod_{t \in \mathcal{C}(r)} M_{rt}(x_t)},$$

recovering equation (17).

For $s \in R$ and $r \in \mathcal{C}(s)$, enforcing the consistency condition $\sum_{x_{s \setminus r}} \tau_s(x_r, x_{s \setminus r}) = \tau_r(x_r)$ gives us

$$\exp\left(\frac{\theta_r(x_r)}{\rho_r}\right) \frac{M_{sr}(x_r)^{\rho_s/\rho_r} \prod_{u \in \mathcal{P}(r) \setminus s} M_{ur}(x_r)^{\rho_u/\rho_r}}{\prod_{t \in \mathcal{C}(r)} M_{rt}(x_t)}$$

$$\propto \sum_{x_{s \setminus r}} \exp\left(\frac{\theta_s(x_s)}{\rho_s}\right) \frac{\prod_{v \in \mathcal{P}(s)} M_{vs}(x_s)^{\rho_v/\rho_s}}{M_{sr}(x_r) \prod_{w \in \mathcal{C}(s) \setminus r} M_{sw}(x_w)}.$$

Rearranging the equation to collect $M_{sr}(x_r)$ on the left hand side and taking the $(1 + \rho_s/\rho_r)$-th root on both sides gives us the update equation (16).

From the derivation above, it is clear that if $\{M_{sr}(x_r)\}$ is a fixed point of the update equation (16), then the collection $\tau$ of pseudomarginals defined by (17) is a stationary point of the Lagrangian (35), since it sets the derivatives of $\mathcal{L}_{\theta,\rho}$ equal to zero.

## E  Additional Simulation Results

In this section, we provide additional plots to better illustrate the observations that we make in Section 5. For convenience, Figures 2(a)–2(d) and Figures 2(i)–2(l) show the same plots as in Figure 1. Figures 2(e)–2(h) show the plots of $(\rho, \log_{10}(\Delta))$ for the Ising models in Figures 2(a)–2(d), and similarly for Figures 2(m)–2(p). Here, $\Delta$ is the final average change of the messages in the sum product algorithm at termination; i.e., either when $\Delta \leq 10^{-10}$ or after 2500 iterations of the algorithm with parallel updates.

For $\rho \leq \rho_{\mathrm{cycle}}$, in which the Bethe variational problem (8) is concave, there is a unique optimal value for the Bethe approximation. The values of $\Delta$ in this region are slightly higher than the convergence threshold, which means sum product has not converged after 2500 iterations, but the final value of $\Delta$ is sufficiently small that the messages have stabilized.

Shortly after $\rho$ becomes larger than $\rho_{\mathrm{cycle}}$, the curve of the Bethe values splits into multiple lines, which indicates that the Bethe objective function has multiple local optima. These lines are evidently distinct local optima since the values of $\Delta$ are at the convergence threshold, which means sum product converges and yields stationary points of the Lagrangian.

In the models with mixed potentials, we observe that for the values of $\rho$ where the multiple local optima begin to emerge, the values of $\Delta$ are significantly higher and sum product does not converge. This behavior is reflected in the presence of the point cloud in the plots of the Bethe values. As noted in Section 5, we suspect that this behavior arises because distinct local optima are initially close together, so messages oscillate between them. For larger values of $\rho$, however, the local optima are sufficiently separated, so sum product converges and there are multiple lines in the graphs of the Bethe values.

(a) $K_5$, mixed, Bethe    (b) $K_5$, attractive, Bethe    (c) $T_9$, mixed, Bethe    (d) $T_9$, attractive, Bethe

(e) $K_5$, mixed, $\log(\Delta)$    (f) $K_5$, attractive, $\log(\Delta)$    (g) $T_9$, mixed, $\log(\Delta)$    (h) $T_9$, attractive, $\log(\Delta)$

(i) $K_{15}$, mixed, Bethe    (j) $K_{15}$, attractive, Bethe    (k) $T_{25}$, mixed, Bethe    (l) $T_{25}$, attractive, Bethe

(m) $K_{15}$, mixed, $\log(\Delta)$    (n) $K_{15}$, attractive, $\log(\Delta)$    (o) $T_{25}$, mixed, $\log(\Delta)$    (p) $T_{25}$, attractive, $\log(\Delta)$

**Figure 2:** Values of the reweighted Bethe approximation and the final $\log_{10}(\Delta)$ as a function of $\rho$.