[Reviews · NeurIPS 2014]

Submitted by Assigned_Reviewer_7

The paper presents various conditions for concavity of reweighted Kikuchi entropy approximation. In particular, for the general region graph, it presents a sufficient condition for concavity that is (somewhat) more general than the condition given in [13] where overcounting number condition is not needed. For two-layer regional graph, a necessary condition is also provided. For pairwise MRF, a graph-based characterization of the concavity condition is given (show to be the same as the single-cycle forest polytope).

These results generalize previously known existing results and could be useful addition to the literature, although I haven't checked the detailed proofs for correctness.

The most glaring error of the paper appears in the background section. By not carefully paying attention to what is "p" in Eq. (4), by the time Eq (6) is written, the authors made the mistake that this is the same "p" in Eq (2). This is not correct. Eq. (6) should read H(\mu) instead of H(p_gamma), and I'm guessing eq. (4) should read H(\tau) instead of H(p) (and likewise in Eq (7)).

Just after 2.2. it is also stated that the set Delta_R is not convex in general. Really? It seems one can easily prove that it is convex, so this statement might also be incorrect. If this is not the case, the authors should say why it is not convex.

Fortunately, from eq. (8) onward, it seems that all these mistakes do not propagate further, and the rest of paper seems OK.

Experiment section: Fig 1 is useful to see the behavior of the approximation around \rho_cycle and also the effects of non-concavity. It is also claimed that \rho_cycle generally gives better approximation than \rho_tree. To do this, one has to present aggregate result over multiple random \gamma, and not just one \gamma as show in Fig 1. I suggest that the authors give a numerical aggregated numbers for accuracy of \rho_tree and \rho_cycle.

Also, for \rho \in forest polytope, the bound can be optimized (lowered) since it is guaranteed to be an upper bound. Can we do anything like that for the single-cycle forest? (It's not clear to me since the bound is not an upperbound, so lower is not necessary better).

Summary: The paper presents various conditions for concavity of reweighted Kikuchi entropy approximation and generalize previously known results. Errors exist in the background section, although after that the paper seems fine, and those errors seem easily fixable.

Submitted by Assigned_Reviewer_14

This paper considers the so-called reweighted version of the Kikuchi variational approximation of the partition function, and provides sufficient conditions under which the variational objective is concave and hence can be optimized more effectively. The conditions are shown to be necessary under additional assumptions. Some experimental results on synthetic problems are shown to confirm the theoretical predictions.

The paper is interesting and well written. It provides the necessary background and it does a good job explaining the results at an intuitive level and placing them into the context of what was previously known. As far as I know the results are novel, and they are a nice contribution to our understanding of variational approximations to the partition function. The results should be of interest for the sub-community of NIPS working on message passing and variational techniques. It's another step forward towards better understanding these techniques and their empirical success. One limitation is that it's a bit unclear if the work has practical implications. For example, it does not introduce any new algorithmic approach to the problem (in the experimental results section, they use the reweighted sum product algorithm of Wainwright et al). It's great that we get a better understanding and we can guarantee convexity and hence convergence, but does the theory provide any a-priori indication on how to set $rho$ for a practitioner? It seems that in practice rho_cycle and rho_tree are fairly close to each other and the corresponding approximations do not differ much.

In the plot in figure 1(d), it seems that the error increases by going from rho_tree to rho_cycle, and it starts underestimating the value of the partition function. Am I reading it correctly? Is that a counterexample that the approximation is not always better and there is no hope it provides an upper bound up to rho_cycle?

Minor:

Line 126: I am curious why it "is not in general convex". Isn't a convex combination of two realizable marginals (points in Delta_R, say realized by tau_1 and tau_2) realizable by taking a convex combination of tau_1 and tau_2 with the same coefficients?
Summary: This paper provides new theoretical results on the concavity of the so-called reweighted version of the Kikuchi variational approximation of the partition function. It's an interesting, well written paper that should be of interest for the variational inference community.

Submitted by Assigned_Reviewer_23

In this paper, the authors analyzed a reweighted Kikuchi approximation of the log partition function. Necessary and sufficient conditions of concavity are established. Also, the single-cycle polytope includes the previously spanning forest polytope which provides a generalization. The paper is well-written and technically sound. The method proposed also leads to a new (and more general) message-passing scheme. I recommend publication.

Comments:

1) While Theorem 1 gives conditions for concavity, it does not give results on what \rho provides good approximation of the log partition function. It would be helpful to provide a discussion on when (or whether) the approximate accuracy is improved.
Summary: The paper provides an analysis of reweighted Kikuchi approximation which leads to a new message-passing schedule. I recommend publication.
Author Feedback
Author rebuttal: We thank the reviewers for their careful reading of our manuscript and their very helpful suggestions. We include point-by-point responses below.

First, thanks to reviewers 7 and 14 for catching a mistake in our introductory material. The R-marginal polytope Delta_R is indeed convex, as pointed out by the reviewers; the reason for replacing Delta_R by the locally consistent superset Delta_R^K is that the latter polytope has polynomially many facets, whereas the number of facets of Delta_R may be exponential.

We also want to clarify an error in our notation as pointed out by reviewer 7. Letting p_tau (respectively, p_mu) denote the uniquely defined distribution with mean parameters tau (respectively, mu), H(p) should be replaced by H(p_tau) in equations (4), (5), and (7), while H(p_gamma) should be replaced by H(p_mu) in equation (6).

Overall, we acknowledge and agree with the reviewers’ assessments about areas where our paper could be strengthened in future work. As noted by reviewers 14 and 23, our paper leaves open the question of how closely the reweighted Kikuchi approximation approximates the log partition function. We do not currently know how to quantify the improvement in moving from rho_tree to rho_cycle, even in the special case discussed in our experiments; as pointed out by reviewer 14, minimizing the variational objective B(theta, rho) over rho_cycle may sometimes produce a value lower than the actual partition function. Therefore, although the arguments used in [20] might be adapted to minimize B(theta, rho) over rho_cycle, the relevance of the resulting bound is unclear in the absence of concrete theoretical results. Quantifying the potential gain of optimizing over rho in the polytope of concavity and understanding when upper- and lower-bounds might be guaranteed are very important avenues for future work.

Finally, we agree with reviewer 7’s assessment that our experimental section would be stronger by reporting aggregate results over multiple gamma, and we intend to include such results in the revision. As commented in lines 403-405, we performed our experiments over multiple random draws of gamma and the results were qualitatively similar.